# Dynamic Behaviour of a Conveyor Belt Considering Non-Uniform Bulk Material Distribution for Speed Control

**Fei Zeng** [1,2,*], **Cheng Yan** [2], **Qing Wu** [3] **and Tao Wang** [1]

[1] Key Laboratory of Metallurgical Equipment and Control Technology of Ministry of Education, Wuhan University of Science and Technology, Wuhan 430081, China; wangtao77@wust.edu.cn

[2] Hubei Key Laboratory of Mechanical Transmission and Manufacturing Engineering, Wuhan University of Science and Technology, Wuhan 430081, China; yancheng@wust.edu.cn

[3] Key Laboratory for Port Handling Technology Ministry of Communication, Wuhan University of Technology, Wuhan 430063, China; wq@whut.edu.cn

\* Correspondence: zengfei@wust.edu.cn; Tel.:+86-15527653091

**Abstract:** For the conveyor belt, variable material flow influences the energy efficiency of the speed control technology significantly. The fluctuation of material flow on the conveyor belt will lead to the detrimental vibrations on both the belt and the conveyor while the conveyor works at certain speeds. In order to improve the model inaccuracy caused by the uniform bulk material flow assumption in the current conveyor belt model, the paper establishes a high-precision dynamic model that can consider speed control of a conveyor belt under non-uniform bulk material transportation. In this dynamic model, a non-uniform bulk material distribution model is firstly proposed based on laser scanning technology. Then, a high-precision longitudinal dynamic model is proposed to investigate the dynamic behavior of a belt conveyor. Considering the micro-units of actual load on a conveyor belt, it can well describe the transient state of the conveyor belt. These models can be used to determine the optimal speed for safety and energy conservation in operation. Experimental results are used to validate the proposed dynamic model for analyzing belt mechanical behavior under non-uniform bulk material distribution on the belt. The results show that the proposed models can be used for optimizing the operating procedures of belt conveyor systems.

**Keywords:** dynamic modelling; belt conveyor; laser scanning; non-uniform bulk material distribution; speed control

## 1. Introduction

Belt conveyor systems are critical equipment for the continuous transportation of bulk materials in many industrial fields [1,2]. With the advantages of long distance, high speed, and large capacity, belt conveyor systems are used widely. A typical belt conveyor system consists of an endless belt for traction and bearing components driven by a high-power redundancy electric roller. Idlers, conveyor belts, and other inertial equipment require less power for smooth transportation than the start-up. In addition, the "big horse pulls a small carriage" phenomenon can occur during the process of conveying bulk materials due to the load changes on the belt and the mode of constant speed operation, as shown in Figure 1. More than 60% of the total energy consumption of bulk material handling and production is spent by the belt conveyor system [3]. If the belt conveyor system starts with no load or operates at nominal speed with unevenly distributed and intermittent bulk materials, a lot of energy is wasted. Therefore, it is necessary to improve the energy efficiency of belt conveyor systems.

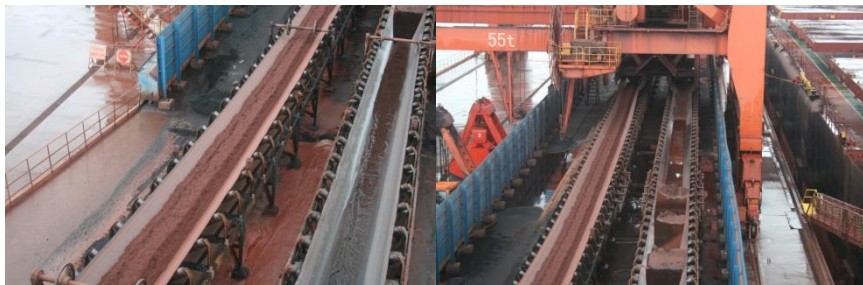

**Figure 1.** The phenomena of unevenly distributed or intermittent bulk materials on the conveyor belt.

A belt conveyor system is a complex electromechanical coupled system. Its energy efficiency can be improved mainly at the equipment level and the operational level. At the equipment level, a variety of techniques of local motor controls have been developed to optimize the operational process of belt conveyor systems, for example, start-up process optimization [4], soft-start device addition [5,6], less motor operation [7,8], asynchronous motor Y-Δ switching [9], and material flow automated control. In addition, a reduced-order model based on a balanced truncation method was proposed in [10], and a predictive controller model based on a reduced-order model performed well in controlling a belt conveyor system. To quantify the energy and cost savings, a motor sequencing controller was installed in [11] to reduce the energy use and to perform initial saving calculations.

Generally, a belt conveyor is a continuous transport machine consists of a conveyor belt and a bearing mechanism. Energy efficiency improvements that occur only at the equipment level may inhibit the production capacity of a bulk terminal or limit certain scenarios since the conveyor belt is usually coupled closely with other mechanical equipment [12]. Operation is another aspect for energy efficiency of belt conveyors. Many researches have explored the effect of optimum control parameter prediction at the operational level. These studies were performed mostly using the global optimization by adjusting the belt speed according to the bulk material flow to reduce energy consumption [13–16]. With the development of the variable frequency technique, the variable speed drive (VSD) strategy has the most potential for energy savings and can be categorized into three categories.

Recent decades have seen the rapid development of intelligent controls [17–19], including fuzzy logic control [20,21], and neural network control [22–24]. In terms of fuzzy logic control, Mazurkiewicz [25] established an expert diagnosis system of belt conveyors to predict the potential fault information in the belt speed condition. Ristić and Jeftenic [26] developed a speed-regulating controller for open-pit mine remote control based on VSD technology. To optimize the speed control strategy, a fuzzy logic method was applied to classify the material feeding rates and the reference speed could be defuzzified based on the linear relationship between belt speed and conveyor capacity [27]. Nevertheless, it is not well adapted to the situation since the relationship is mainly by experience or trial and error. Based on the system optimum loading rate principle, Jeftenić et al. [28] deduced an optimal belt velocity by making the cross-section of the material flow reach the optimal loading rate of the belt conveyor. The remote control of belt conveyors for energy savings in open pits has been realized through direct torque control (DTC). To accurately predict the optimum belt speed, RISTIĆ et al. [29] established the relationships between the belt speed and material instantaneous flow rate. Still, the energy consumption mechanism of a belt conveyor has not been reported yet.

Presently, optimal control is a well-accepted approach in energy optimization field [30–32]. Based on the practical engineering problem, Lodewijks [33] deduced a speed control strategy that considered the parameters of friction force, power consumption, number of bags per hour and the interval time between two adjacent bags to reduce the power consumption of the baggage transport system. Mathematically, the above coordination is usually formulated as a kind of optimization problem. Typically, an accurate energy model is one of the key optimization objectives of belt conveyor's energy optimization problem. Based on the International Organization for Standardization (ISO) 5048, an energy model of belt conveyors was proposed for calculation of the drive power [34].

Then, this model can be used as the optimization objective with other operation constraints to assess the optimal scheduling of conveyor belts in a coal power plant [35]. With the consideration of time-of-use (TOU) tariff, load shifting was achieved by operation efficiency optimization. In order to improve the practicability, a closed-loop model predictive control (MPC) methodology was proposed to deal with the disturbances arising from coal consumption forecasting and the feed rate [36]. Furthermore, Zhang and Mao [37] took conveying systems and crushers as a whole for energy efficiency optimization and firstly investigated the energy models of belt conveyors and ring hammer crushers.

It should be noted that to a large extent, the dynamic characteristics determine the working performance [38]. However, both energy improvement methods at equipment level and operational level did not consider the dynamic characteristics of a belt conveyor in transient operation. For example, detrimental vibrations may occur on the belt and conveyor structures when the material flow fluctuates. That is because the dynamic characteristics of a fabric conveyor belt is nonlinear and viscoelastic. When speed adjustment is not properly controlled on the belt, it will cause potential risks, such as belt slipping around pulley and belt tearing at the splicing area, especially when the speed changes dramatically [39]. Based on Deutsches Institut für Normung (=German Industry Standard, DIN) 22101, Lauhoff [40] questioned the appropriateness of speed control for energy savings because the nonlinear and viscoelastic dynamic characteristics of the conveyor belts were not considered.

The speed control strategy must fulfil the reliability and safety requirements of the belt conveyor system. He et al. [41–43] proposed the Estimation–Calculation–Optimization (ECO) method to adjust the control parameters of the belt conveyor system after dynamic characteristics analysis. It could greatly improve the reliability of the speed control system of the belt conveyor. Štatkić [44] provided a reliability assessment of a single motor drive on a belt conveyor station with a changeable structure of frequency converter power modules. Nevertheless, their approach did not consider actual non-uniform material distribution either.

Furthermore, today's conveyor belts have been built with longer transmission distance, up to tens of kilometers. For such a long conveyor belt, complex characteristics will be always expected, such as nonlinear, hysteresis, creep, and relaxation. It becomes even more complicated due to the interactions of the belt and the bulk material. At present, a variety of theoretical and numerical methods have been developed to simulate the dynamic characteristics of this long belt [45–47]. The theoretical calculation of longitudinal belt characteristics only considers a uniform bulk material distribution. However, the non-uniform bulk material distribution has been rarely considered for optimal speed control. In the previous study [48], we presented a noncontact measurement system and a bulk material flow calculation method using laser scanning technology. It could contribute to establishing a mechanical model of a loaded belt based on more accurate bulk material flow data. Considering the previous discussion, the main purpose of this paper is to propose a high-precision dynamic model intended to determine the dynamic characteristics of a conveyor belt under non-uniform bulk material distribution. In addition, a method for measuring the actual bulk material distribution using laser scanning technology is presented.

In this paper, an experimental facility for bulk material distribution measurement based on laser scanning technology is designed and constructed firstly. After acquiring the bulk material cross-sections in real-time, a non-uniform bulk material distribution model is proposed. Due to the bulk material on the conveyor belt is discretized into a series of micro-units, this distribution model can describe the actual load distribution on a conveyor belt. Based on this, a high-precision longitudinal dynamic model to investigate the dynamic behavior of a belt conveyor is investigated. We represent the experimental verification results for the non-uniform bulk material distribution model and establish the discrete simulation model of the belt conveyor system using MATLAB software. Then, we present a simulation test on the different belt tensions, accelerations and tensioning device displacements at the head and tail of the belt conveyor system. The discussion on the influence of the non-uniform bulk material distribution and starting time on the dynamic characteristics of the belt conveyor is proposed,

which contribute to the establishment of speed control constraints for developing a control strategy for practical application.

The layout of the rest of the paper is as follows: Section 2 introduces the laser scanner on-line measurement system and describes the theoretical determination of the non-uniform bulk material distribution on the conveyor belt with the data collected by the laser scanner online measurement system, and then formulates an improved mathematical model considering the non-uniform bulk material distribution to determine the dynamic characteristics of the conveyor belt in Section 3. Section 4 presents the numerical simulation study of a belt conveyor dynamic behavior with non-uniform bulk material distribution, and Section 5 discusses the results. Section 6 concludes the paper.

## 2. Actual Non-Uniform Bulk Material Distribution Measurement Methodology

### 2.1. Problem Formulation

A high-precision dynamic model is the basis of the dynamic analysis of the belt conveyor. At present, the commonly used dynamic models of belt conveyors include partial differential equations based on continuous systems and one- or two-dimensional vibration finite element models based on discretized systems. However, the above models for the longitudinal belt dynamic characteristics analysis typically assume a uniform distribution of bulk material on the belt. The belt dynamic characteristics is more complicated because of the interaction of the bulk material and the belt. The complexity is increased when considering the effect of the actual non-uniform bulk material distribution on conveyor belt tension during the speed changing process. This complex interaction makes it difficult to measure and predict the belt tension change accurately, which may lead to slip, wear, or tear of the belt. Therefore, the reliability and safety of belt conveyor systems can be effectively assured if belt tension can be predicted accurately, and control parameters can be selected reasonably when energy-saving controls are implemented.

In this part, a non-uniform bulk material distribution model based on bulk material flow measurement data from a laser scanner is proposed. The data are used to establish the balance equation of the conveyor belt when considering the effect of the bulk material distribution. A longitudinal dynamic model of the belt conveyor system is established by using the finite element method (FEM). Finally, the tension change law of the belt, considering the non-uniform bulk material distribution, will be drawn from the simulation analysis. This provides a theoretical basis for establishing an energy-saving control strategy for a belt conveyor system.

### 2.2. Measurement Facility

An experimental facility for bulk material distribution measurement is designed and constructed, as shown in Figure 2. It is designed to experimentally measure bulk material cross-sections in real-time and can be used to study the mechanical properties of a belt under actual material distribution conditions. The facility conforms to DIN 22101 (1982) "Type and Basic Parameters of Belt conveyors", which can simulate a real process of continuous transportation of bulk material in laboratory conditions. It consists of a variable frequency controller which can adjust the belt speeds varying from 0.2 to 4 m/s. Due to the limited size of the laboratory space, the longitudinal length, horizontal width, and height of the facility are designed as 3500 mm, 760 mm, and 400 mm, respectively. It is a trough-type belt conveyor with three idlers, and the idler roll set spacing is 700 mm. The height of the facility is adjustable from 400 mm to 1338 mm to change the inclination angle of the belt conveyor system. The parameters of the experimental facility are listed in Table 1. A (7000 × 500 × 11 mm) rubber canvas belt is used to test the non-shrinkage of bulk materials. The electrical control components of the facility include a squirrel cage asynchronous motor, power switch box and frequency converter. The parameters of the frequency converter can be modified to realize open-loop control functions (starting, positive or negative rotating, etc.) or closed-loop control functions.

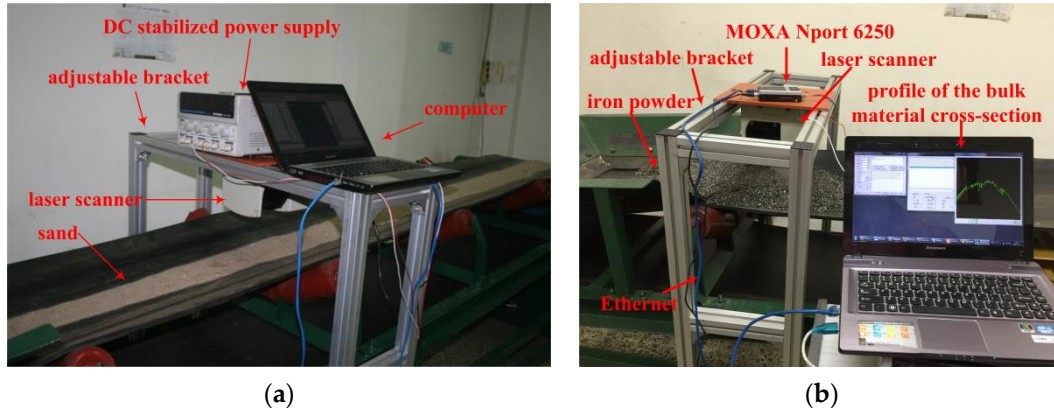

**Figure 2.** Experimental facility for measuring bulk material distribution. (**a**) Sand cross-sections measurement in real-time; (**b**) the fitting of iron powder cross-section in real-time.

In the bulk material distribution measurement system, a laser scanner (Sick LMS291-S05, made in Germany) is installed on the adjustable bracket and located above the conveyor belt to scan the profile of the bulk material cross-section orthographically. The parameters of angular resolution, scanning angle, resolution/typical measurement accuracy and maximum range are configured respectively by using the supplied LMSIBS configuration software. Then the laser scanning points are transmitted to the computer via Ethernet by using a MOXA Nport 6250. In addition, a belt speed monitor (SEW ES1T (OG 72 DN 1024 TTL)) is mounted on the central axis of the driving wheel to collect the belt speed synchronously. After the belt speed is detected by the speed monitor, the microcontroller STM32 records it and sends it to the computer in time through a GSM & GPRS module GTM900C. Accordingly, a three-dimensional point cloud of the material cross-section moving on the belt conveyor can be acquired in real-time. Using the software developed in our laboratory, the profiles and areas of the bulk materials cross section are obtained in real-time.

**Table 1.** Parameters of the experimental facility.

| Parameter Description | Value | Parameter Description | Value |
|---|---|---|---|
| Facility longitudinal length, mm | 3500 | Belt length, mm | 7000 |
| Facility horizontal width, mm | 760 | Belt width, mm | 500 |
| Facility height, mm | 400 | Belt thickness, mm | 11 |
| Facility height range, mm | 400~1338 (adjustable) | Drive motor | Shanghai shenli yvf2-90l-4 |
| Idler roll set spacing, mm | 700 | Reduction ratio | 1:29 |
| Inclination angle, ° | $\delta \le 8°$ | Large and small gear ratio | 24:16 |
| Conveyor Belt | Rubber canvas ordinary | Frequency converter | ABB:ACS550-01-05A4-4 |

### 2.3. Non-Uniform Bulk Material Distribution Model

The non-uniform bulk material distribution affects the dynamic performance of the system. Belt tear and breaking accidents can occur when the belt encounters unsteady working conditions, such as starting and braking, since the dynamic characteristics of the belt under a non-uniform bulk material distribution are more complex. Moreover, the type of material, the conveying process requirements, the equipment layout and other factors often cause non-uniform bulk material distribution in actual operation, as shown in Figure 3, it makes the dynamic characteristics of the belt even more complex.

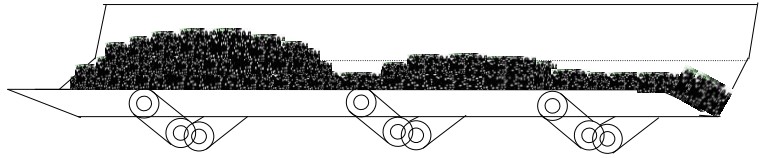

**Figure 3.** The non-uniform bulk material distribution in actual operation.

To describe the non-uniform bulk material distribution on a conveyor belt, we discretize the bulk material on the conveyor belt into a series of micro-units according to the travel distance *l* in unit time *T*. Assumed *T* contains *T* seconds, *t* is the *t*-th unit of time between time *(t-1)T* and time *tT*, *k* is the number of material flow cross-section areas per unit time. Accordingly, the non-uniform bulk material distribution $q_G(t) = \{q(t_1), q(t_2), \ldots \ldots, q(t_k)\}$, which denotes the collection of *k* sampling bulk material loads in unit time *T* between time *(t-1)T* and time *tT*, can be represented as Figure 4. Due to the scanning frequency which the laser scanner can capture frames of cross section per second is $f_{speed}$, *k* is obtained by $k = T \cdot f_{speed.}$. Then $t_k$ means the *k* frame of *t*-th unit of time. Compared with the *tT* (*t* is from 1~*t*), the $t_k$ (*t* is from 1~*t*, *k* is from 1~*k*) can much better reflect the original time series. Based on the model of cross-sectional area of the bulk material proposed in [49], the bulk material distribution model can be presented.

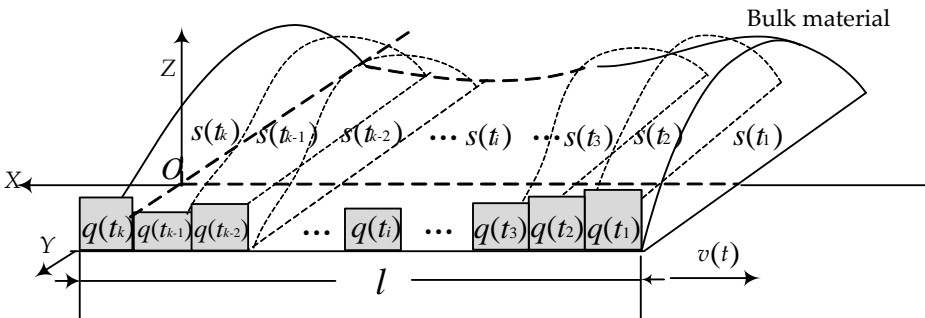

**Figure 4.** Schematic diagram of the micro-units of a non-uniform bulk material distribution on the conveyor belt.

In Figure 4, the belt moves in the negative direction of the *x*-axis. $s(t) = \{s(t_1), s(t_2) \ldots, s(t_k)\}$ is the collection of cross-sectional areas of material flow calculated in unit time *T* between time *(t-1)T* and time *tT*. $v(t) = \{v(t_1), v(t_2) \ldots, v(t_k)\}$ is the collection of belt speed in unit time *T* between time *(t-1)T* and time *tT*. Therefore, the bulk material distribution $q_G(t)$ (kg/s) in unit time *T* between time *(t-1)T* and time *tT* could be represented as

$$q_G(t) = \left\{ \frac{\rho}{f_{speed}-1}s(t_1)v(t_1), \ \frac{\rho}{f_{speed}-1}s(t_2)v(t_2), \cdots \cdots, \ \frac{\rho}{f_{speed}-1}s(t_k)v(t_k) \right\} \tag{1}$$

where $\rho$ is the bulk density, $\{v(t_1), v(t_2) \ldots, v(t_k)\}$ is the successive instantaneous speed of the belt in unit time *T* measured by the speed monitor, (m/s), $\{s(t_1), s(t_2) \ldots, s(t_k)\}$ is the successive cross-sectional area of material flow in the *i* frame in unit time *T* (m$^2$).

## 3. The Longitudinal Dynamics Model of the Belt Conveyor with Non-Uniform Bulk Material Distribution

The belt conveyor is a complex nonlinear, transient and coupled system. The implementation of safety control of the belt conveyor needs accurately predicted dynamic characteristics of the belt when the conveyor works under unsteady conditions. Therefore, a precise longitudinal dynamic model of the belt conveyor is established using the finite element method (FEM) combined with the non-uniform bulk material distribution model. Figure 5 shows the transmission principle of the belt conveyor. The belt conveyor consists of two driven rollers at the head, a single return drum at the tail and a tensions device. The deformation, speed and acceleration along the elastic conveyor belt are different at a given time. These kinematic parameters depend on the position of the conveyor belt and time.

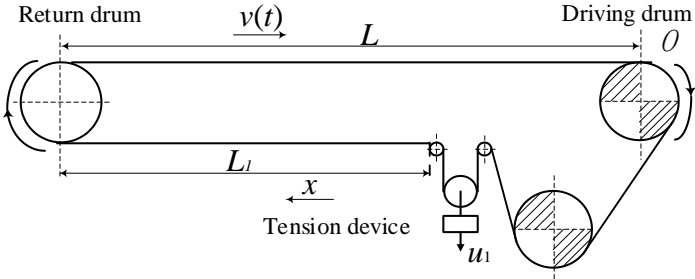

**Figure 5.** The transmission principle of the belt conveyor.

### 3.1. Discrete Model of the Belt Conveyor System

Firstly, the belt is discretized by the finite element method. Different element sizes are set to model the carrying section and the return section of the belt. The carrying section of the belt is discretized in a very high resolution to model the changes of dynamic tension. Conversely, the return section of the belt is roughly discretized to reduce computational complexity. The Kelvin–Vogit model is adopted to represent the characteristic of the conveyor belt. Then, a discrete closed-loop model of the belt conveyor system can be formed by connecting the elements successively (Figure 6).

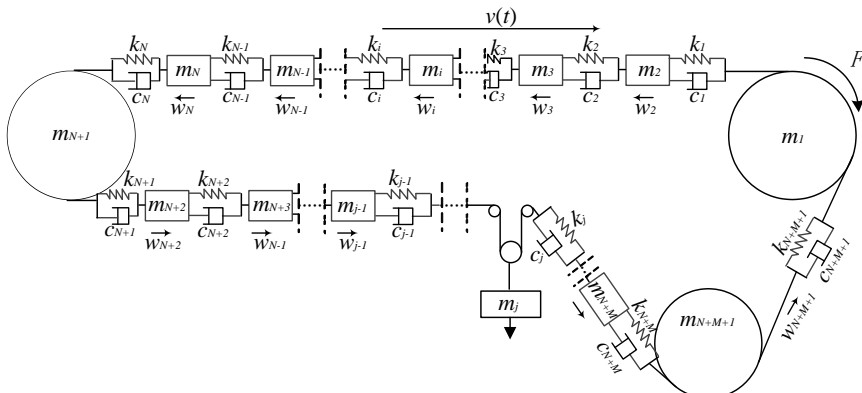

**Figure 6.** A discrete closed-loop model of the belt conveyor system.

In Figure 6, the belt is discretized into $N+1$ elements along the opposite running direction of the conveyor belt. The driven roller of the head is the No. 1 and the return drum at the tail is the No. $N + 1$. The length of the element of the belt at the carrying section is set to $\bar{v}(t) \cdot T$, and $N \approx \text{int}(L/\bar{v}(t) \cdot T)$, where $L$ is the length of the belt conveyor, $\bar{v}(t)$ is the mean value of the transient speed $v(t)$ in the time $t \cdot T$. The length of the element of the belt at return section is set to a multiple of $\bar{v}(t) \cdot T$. Similarly, the belt at return section is divided into $M$ elements starting from the tail (No. $N + 2$) to the end driven roller (No. $N + M + 1$), and $M \approx \text{int}(L/r \cdot \bar{v}(t) \cdot T)$, where $r$ is the constant coefficient.

### 3.2. Element Mass

The belt conveyor is discretized into $N + M + 1$ elements according to the travel distance, of which $N + 1$ elements within unit time $T$ at the bearing section and $M$ elements within time $rT$ at the return section. Considering the non-uniform bulk material distribution, the $m_t$ $(x(tT), tT)$ at the bearing section of the belt conveyor is expressed as

$$m_t(x(tT), tT) = \frac{1}{f_{speed} - 1} \left[ (q_B + q_{RO}) \sum_{i=1}^{k} v(t_i) + \rho \sum_{i=1}^{k} s(t_i)v(t_i) \right] \tag{2}$$

where $q_B$ is the mass of the conveyor belt per unit length at the bearing section, (kg/m), $q_{RO}$ is the rotating mass of the roller per unit length at the bearing section, (kg/m), and $t$ is the $t$th infinitesimal element, $t$ values from 1 to $N + 1$.

On the other hand, the $m_t(x(t{\cdot}rT), t{\cdot}rT)$ at the return section of the conveyor belt is formulated as

$$m_t(x(t \cdot rT), t \cdot rT) = \frac{1}{f_{speed} - 1}(q_B + q_{RU}) \sum_{i=1}^{rk} v(t_i) \tag{3}$$

where $q_B$ is the mass of the conveyor belt per unit length at the return section, (kg/m), $q_{RU}$ is the rotating mass of the roller per unit length at the return section, and $t$ is the $t$th infinitesimal element, (kg/m), $t$ values from $N + 2$ to $N + M + 1$.

### 3.3. Force Equilibrium Equation of the Conveyor Belt

#### 3.3.1. At the Bearing Section

As the conveyor belt travels along the positive direction, the displacement $x(tT)$ of one element of the belt at the bearing section at time $tT$ can be expressed as

$$x(tT) = x'(tT) + \frac{1}{f_{speed} - 1} \sum_{t=1}^{t} \sum_{i=1}^{k} v(t_i) \tag{4}$$

where $x'(tT)$ is the elastic displacement of the conveyor belt at time $tT$, $t$ is the $t$th infinitesimal element, $t$ values from 1 to $N + 1$.

The force equilibrium equation of the $t$th infinitesimal element at the displacement $x(tT)$ at time $tT$ is formulated as

$$S_t(x(tT), tT) - S_t(x((t+1)T), (t+1)T) = m_t(x(tT), tT)\ddot{x}(tT) + w_t(x(tT), tT) \tag{5}$$

where $m_t(x(tT), tT)$ is the $t$th infinitesimal element mass of the belt conveyor, $S_t(x(tT), tT)$ is the belt tension of the $t$th infinitesimal element at the displacement $x$ at time $tT$, can be calculated by

$$S_t(x(tT), tT) = k_{tT}(x(tT) - x((t+1)T)) + c_{tT}(\dot{x}(tT) - \dot{x}((t+1)T)) \tag{6}$$

where $k_{tT}$ is the stiffness coefficient of the belt at the displacement $x(tT)$ at time $tT$ of $t$th infinitesimal element, which can be determined by

$$k_{tT} = \frac{(f_{speed} - 1)E'_B B}{\sum_{i=1}^{k} v(tT)} \tag{7}$$

where $B$ is bandwidth, (mm), $E'_B$ is the equivalent elastic modulus of the conveyor belt, which is used to describe the influence on characteristics of belt by the deformed belt between adjacent idlers due to the weight of the conveyor belt and bulk material distribution (Figure 7).

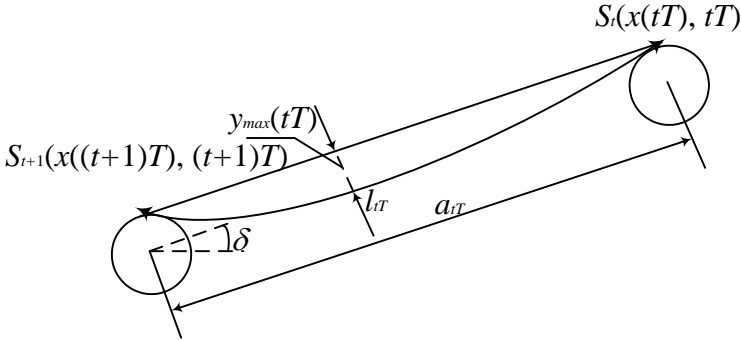

**Figure 7.** An overhang of the belt between adjacent idlers.

As shown in Figure 7, we assume $a_{tT}$ is the idler spacing at the displacement $x(tT)$ at time $tT$, (m), $\delta$ is the conveying inclination of the belt, (°), $l_{tT}$ is the arc length at the displacement $x$, (m), $q_B$ is the mass of the conveyor belt per unit length (kg/m), and $S_t(x(tT), tT)$ is the belt tension of $t$th infinitesimal element at the displacement $x(tT)$ at time $tT$.

Then $E'_B$ can be determined by

$$E'_B = \cfrac{E_B}{1 + \cfrac{(q_B + \frac{\rho}{f_{speed}-1}\sum\limits_{i=1}^{k} s(t_i)v(t_i))^2 g^2 a_{tT}^2 \cos^2 \delta}{12 S_t^3(x(tT),tT)} E_B B} \tag{8}$$

where $E_B$ is the elastic modulus of conveyor belt which can be measured experimentally, and the calculation formula is

$$E_B = \frac{\Delta Q \times L_0}{\Delta L} \tag{9}$$

where $\Delta Q$ is the load difference on the unit width of the specimen, (N/mm), $\Delta L$ is the elongation of the specimen, and $L_0$ (mm) is the original length of the specimen, (mm).

Assume that $c_{tT}$ is the composite viscosity coefficient for materials and idlers at the displacement $x(tT)$ at time $tT$ of $t$th infinitesimal element, and it can be determined by

$$c_{tT} = \frac{(f_{speed} - 1)E'_B B \tau}{\sum\limits_{i=1}^{k} v(tT)} \tag{10}$$

where $\tau$ is the rheological constant, usually from 0.8 to 1. $w_t(x(tT), tT)$ is the main resistance of $t$th infinitesimal element at the displacement $x(tT)$ at time $tT$ of the belt at the bearing section, which is calculated by

$$\begin{aligned} w_t(x(tT), tT) &= \frac{f_{tT} q_{RO} g}{f_{speed}-1}(C_{v0} + C_v \frac{\partial x(tT)}{\partial t})\sum_{i=1}^{k} v(t_i) + \frac{g \sin \delta_{tT}}{f_{speed}-1}[q_B \sum_{i=1}^{k} v(t_i) + \rho \sum_{i=1}^{k} s(t_i)v(t_i)] \\ &+ \frac{f_{tT} g \cos \delta_{tT}}{f_{speed}-1}(C_{v0} + C_v \frac{\partial x(tT)}{\partial t})[q_B \sum_{i=1}^{k} v(t_i) + \rho \sum_{i=1}^{k} s(t_i)v(t_i)] \end{aligned} \tag{11}$$

where $C_{vo}$ is the coefficient independent of the belt speed, $C'_v$ is the coefficient related to the belt speed, and $f_{tT}$ is the dynamic resistance coefficient at the displacement $x$ at time $tT$ and can be determined by

$$f_{tT} = f_0(C_{vo} + C'_v v(t_i)) \tag{12}$$

where $f_0$ is the coefficient of static friction between the belt and the material and usually equal to 0.03, $C_{vo}$ is the coefficient independent of the belt speed, and $C'_v$ is the coefficient dependent of the belt speed.

### 3.3.2. At the Return Section

The displacement $x(t \cdot rT)$ of one element of the belt at the return section at time $t \cdot rT$ can be expressed as

$$x(t \cdot rT) = x'(t \cdot rT) + \frac{1}{f_{speed} - 1} \sum_{t=1}^{t} \sum_{i=1}^{rk} v(t_i) \tag{13}$$

where $x'(t \cdot rT)$ is the elastic displacement of the conveyor belt at time $t \cdot rT$, $t$ is the $t$th infinitesimal element, $t$ values from $N + 2$ to $N + M + 1$.

The force equilibrium equation of the $t$th infinitesimal element at the displacement $x(t \cdot rT)$ at time $t \cdot rT$ is formulated as

$$S_t(x(t \cdot rT), t \cdot rT) - S_t(x((t+1)rT), (t+1)rT) = m_t(x(t \cdot rT), t \cdot rT)\ddot{x}(t \cdot rT) + w_t(x(t \cdot rT), t \cdot rT) \tag{14}$$

where $m_t (x(t \cdot rT), t \cdot rT)$ is the $t$th infinitesimal element mass of the belt conveyor at the return section, $S_t(x(t \cdot rT), t \cdot rT)$ is the belt tension of the $t$th infinitesimal element at the displacement $x(t \cdot rT)$ at time $t \cdot rT$, can be calculated by

$$S_t(x(t \cdot rT), t \cdot rT) = k_{t \cdot rT}(x(t \cdot rT) - x((t+1)rT)) + c_{t \cdot rT}(\dot{x}(t \cdot rT) - \dot{x}((t+1)rT)) \tag{15}$$

where $k_{t \cdot rT}$ is the stiffness coefficient of the belt at the displacement $x(t \cdot rT)$ at time $t \cdot rT$ of $t$th infinitesimal element at return section, which can be determined by

$$k_{t \cdot rT} = \frac{(f_{speed} - 1)E'_B B}{\sum\limits_{i=1}^{rk} v(t_i)} \tag{16}$$

where $E'_B$ can be determined by

$$E'_B = \frac{E_B}{1 + \frac{q_B^2 g^2 a_{t \cdot rT}^2 \cos^2 \delta}{12 S_t^3(x(t \cdot rT), t \cdot rT)} E_B B} \tag{17}$$

where $a_{t \cdot rT}$ is the idler spacing of $t$th infinitesimal element at return section at time $t \cdot rT$.

Assume that $c_{t \cdot rT}$ is the composite viscosity coefficient for materials and idlers at the displacement $x(t \cdot rT)$ at time $t \cdot rT$ of $t$th infinitesimal element at return section, and it can be determined by

$$c_{t \cdot rT} = \frac{(f_{speed} - 1)E'_B B \tau}{\sum\limits_{i=1}^{rk} v(t_i)} \tag{18}$$

At the return section, $w_t(x(t \cdot rT), t \cdot rT)$ is

$$w_t(x(t \cdot rT), t \cdot rT) = \frac{f_{t \cdot rT}(q_{RU} + q_B \cos \delta)g}{f_{speed} - 1}\left(C_{v0} + C_v \frac{\partial x(t \cdot rT)}{\partial t}\right)\sum_{i=1}^{rk} v(t_i) - \frac{q_B g \sin \delta}{f_{speed} - 1} \cdot \sum_{i=1}^{rk} v(t_i) \tag{19}$$

where $f_{t \cdot rT}$ is the dynamic resistance coefficient at time $t \cdot rT$ at return section and can be determined by Equation (12).

### 3.4. Dynamics Model of the Belt Conveyor System

The belt conveyor system is discretized into $N + M + 1$ elements. The first driver motor is numbered as the first element. Assuming that the conveyor belt does not slip on the roller, the equivalent mass

of the drive motor at the straight direction can be put in the mass of the first element. Similarly, the equivalent mass of the second driver motor can be put in the mass of the $N + M + 1$ element.

Based on Newton's Second Law, the dynamic model of a belt conveyor can be described as

$$
\begin{cases}
(m_1 + \frac{J_1}{R_1^2})\ddot{x}_1 + (k_1 + k_2)x_1 - k_1 x_2 - k_n x_n + (c_1 + c_n)\dot{x}_1 - c_1\dot{x}_1 + c_n\dot{x}_n = F_{d1} + F_{d2} - w_1 \\
\qquad m_2\ddot{x}_2 + (k_2 + k_1)x_2 - k_1 x_1 - k_2 x_3 + (c_2 + c_1)\dot{x}_2 - c_1\dot{x}_3 = -w_2 \\
\qquad\qquad\qquad\qquad \vdots \\
\qquad m_j\ddot{x}_j + (k_j + k_{j-1})x_j - k_{j-1}x_{j-1} - k_j x_{j+1} + (c_j + c_{j-1})\dot{x}_j - c_{j-1}\dot{x}_{j+1} = -w_j \\
\qquad\qquad\qquad\qquad \vdots \\
(m_n + \frac{J_2}{R_2^2})\ddot{x}_n + (k_n + k_{n-1})x_n - k_{n-1}x_{n-1} - k_n x_{n+1} + (c_n + c_{n-1})\dot{x}_n - c_{n-1}\dot{x}_{n+1} = -w_n
\end{cases}
\tag{20}
$$

where $J_1$ and $J_2$ are the rotational inertias of the two driven rollers, $R_1$ and $R_2$ are the radius of the two driven rollers, $n$ is the total number of the elements, which is equal to $N+M+1$, $\{m_1, m_2, \ldots m_j \ldots m_n\}$ are the mass of the {1th, 2th, … , jth, … nth} element respectively, which can be calculated by Equations (2) or (3). $\{k_1, k_2, \ldots k_j \ldots k_n\}$, $\{c_1, c_2, \ldots c_j \ldots c_n\}$ and $\{w_1, w_2, \ldots w_j \ldots w_n\}$ are the stiffness coefficients, the composite viscosity coefficients and the main resistances of the {1th, 2th, … , jth, … nth} element, respectively; $\{x_1, x_2, \ldots x_j \ldots x_n\}$, $\{\dot{x}_1, \dot{x}_2, \ldots \dot{x}_j \ldots \dot{x}_n\}$, $\{\ddot{x}_1, \ddot{x}_2, \ldots \ddot{x}_j \ldots \ddot{x}_n\}$ are the displacement, velocity, and acceleration of the linear motion of the {1th, 2th, … , jth, … nth} element of the conveyor belt respectively, $F_{d1}$ and $F_{d2}$ are the equivalent driving forces acts on driving rollers.

Therefore, the matrix form of Equation (20) is

$$
M\ddot{X} + C\dot{X} + KX = F(t) \tag{21}
$$

where $M$ is a matrix of mass, $C$ is a matrix of damping factor, $K$ is a matrix of spring factor, $F(t)$ is a vector of external forces, $X$ is a vector of element displacement, $\dot{X}$ is a vector of element velocity, and $\ddot{X}$ is a vector of element acceleration.

## 4. Simulation Conditions and Procedures

### 4.1. Measurement of the Bulk Material Distribution

The traditional dynamic model of belt conveyor assumes that the bulk materials are uniformly distributed, without considering the influence of the non-uniform distribution of bulk materials. The non-uniform bulk material distribution model is obtained by integrating the instantaneous bulk material flow cross section, and the accuracy is higher than that of the traditional uniform distribution model.

Experimental measurements have been used to verify the correctness and accuracy of the measurement system of the non-uniform bulk material distribution (Figure 8). The materials for the measurement is iron ore powder (bulk density $\rho$, 2980 kg/m$^3$) which provided by Wuhan Iron & Steel (group) Corp. To measure the distribution of the iron ore powder on the moving belt, constant speeds of 0.5 m/s, 1.0 m/s, and 1.5 m/s were selected as suitable values, which were controlled by using the frequency converter in the control cabinet. A spherical cap was used to calibrate the spatial coordinates captured by the measurement system. After that, the perpendicular distance and horizontal width of the adjustable bracket were set 583 mm and 820 mm respectively. The resolution of the laser scanner was set 10 mm/±35 mm over a typical range of 30 m. Next, experiments were performed as the iron ore powder was poured on the belt artificially in three flow rates of 2.971 kg/s (1 L/s), 8.911 kg/s (3 L/s), and 14.853 kg/s (5 L/s) from the guild chute. Standard volumes of 1 L, 3 L, 5 L were used to verify the volumes measured by the measurement system. The triangular area accumulation method [48] was used to calculate the cross-sectional area of iron ore powder. Then, the measured data of instantaneous

iron ore powder distribution with a flow rate of 8.911 kg/s (3L/s) at a stable speed of 1.0 m/s are shown in Figure 9.

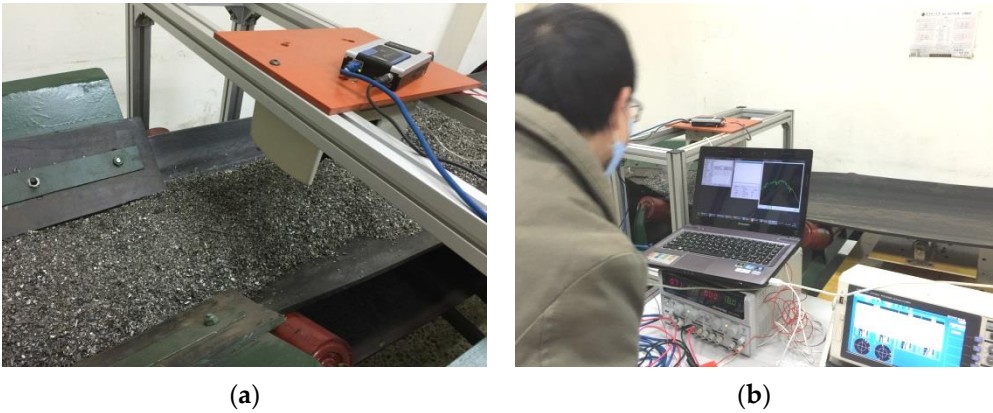

(**a**)                  (**b**)

**Figure 8.** Experimental measurements of the non-uniform iron ore powder distribution. (**a**) Iron ore powder in standard volume was poured on the belt artificially. (**b**) The fitting of iron powder cross-section in real-time.

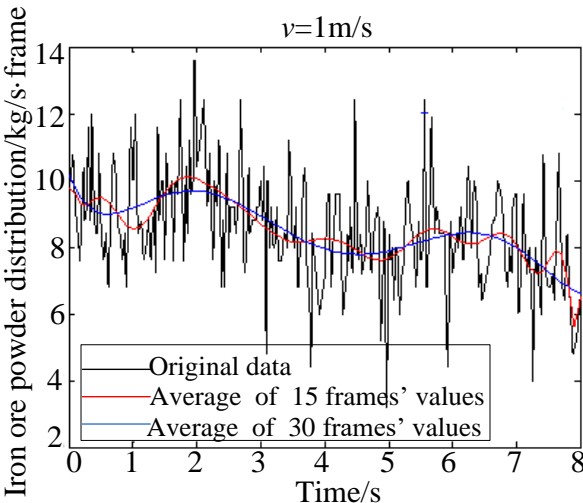

**Figure 9.** The measured data of instantaneous non-uniform iron ore powder distribution with a flow rate of 8.911 kg/s (3L/s) at a stable speed of 1.0 m/s.

The original data curve in Figure 9 shows that the transient iron ore powder distribution fluctuates violently. Due to the high resolution of the laser scanner, the non-uniform bulk material distribution model can better reflect the real state of bulk material transportation. However, the violently fluctuation of original data may affect the measurement of real logistics morphology. Therefore, a mean filter is adopted to smooth the original data. Through experimentation, the average of every 30 frames is smoother than every 15 frames. For verification, 9 sets of measurements had been designed to obtain non-uniform iron ore powder distribution. Every 8 measurement values from 10 measurements corresponding to the same level standard flow rate of iron ore power were recorded after removing the maximum and minimum values. Table 2 shows the repeated measurements corresponding to standard volumes at three speeds of the belt and their repeatability. The average error between the laser measurement and the artificial measurement is within 3%. The repeatability and correlation (*RPT*) are more than 98%. In addition, the variation coefficient (CV) of the measurement value at different speeds is less than 2%, which means that the belt speed has a small influence on the measurement results. In the subsequent dynamic simulation test, the non-uniform bulk material distributions on

belt, which are used for establishing the discrete model of the belt conveyor system, can be obtained from this measurement system.

**Table 2.** Measured values of the iron ore powder distribution based on laser scanning and their repeatability.

| Test NO. | Belt Speed (m/s) | Artificial Flow Rate (L/s) | Distribution Calculated by the Accumulation of the Triangular Area (dm$^3$) | | | | | | | | | | | |
|---|---|---|---|---|---|---|---|---|---|---|---|---|---|---|
| | | | 1 | 2 | 3 | 4 | 5 | 6 | 7 | 8 | *Mean* | *Stdev* | *CV* | *RPT* |
| 1 | 0.5 | 1 | 0.985 | 1.015 | 1.016 | 0.978 | 1.003 | 0.992 | 0.968 | 0.991 | 0.994 | 0.017 | 0.017 | 0.983 |
| 2 | 0.5 | 3 | 2.989 | 3.101 | 2.954 | 2.963 | 2.987 | 3.013 | 2.981 | 3.022 | 3.001 | 0.046 | 0.015 | 0.985 |
| 3 | 0.5 | 5 | 5.017 | 4.989 | 4.993 | 5.003 | 4.995 | 4.899 | 5.100 | 5.002 | 5.000 | 0.054 | 0.011 | 0.989 |
| 4 | 1 | 1 | 1.019 | 0.997 | 0.988 | 1.026 | 1.003 | 0.996 | 1.011 | 0.987 | 1.003 | 0.014 | 0.014 | 0.986 |
| 5 | 1 | 3 | 2.984 | 3.063 | 3.013 | 2.999 | 2.896 | 2.989 | 2.986 | 3.001 | 2.991 | 0.046 | 0.015 | 0.985 |
| 6 | 1 | 5 | 4.898 | 5.016 | 4.895 | 5.012 | 5.008 | 4.995 | 4.986 | 4.929 | 4.967 | 0.052 | 0.010 | 0.990 |
| 7 | 1.5 | 1 | 1.024 | 0.989 | 1.003 | 0.989 | 0.976 | 0.989 | 0.978 | 0.997 | 0.993 | 0.015 | 0.015 | 0.985 |
| 8 | 1.5 | 3 | 2.998 | 3.021 | 2.982 | 2.978 | 3.104 | 2.973 | 2.990 | 2.976 | 3.003 | 0.044 | 0.015 | 0.985 |
| 9 | 1.5 | 5 | 4.983 | 4.889 | 5.020 | 4.915 | 4.979 | 5.042 | 4.987 | 5.037 | 4.982 | 0.055 | 0.011 | 0.989 |

*4.2. Dynamic Simulation Analysis*

Using the MATLAB simulation platform, a discrete model of the conveyor is established and the changing laws of belt tension, speed, and displacement under the non-uniform bulk material distribution are obtained through a simulation test. All the simulation cases are coded with M files in MATAB R2010b. The user interface is designed by the graphical user interface (GUI) toolbox of MATAB. When the non-uniform bulk material distribution measured by the experimental facility mentioned in Section 2.2, the real and simulated parameters of the belt conveyor are input to the GUI, the matrix of *M*, *C*, *K* in Equation (21) can be built. After a soft start control is applied on the belt conveyor, the ode 45 function in MATAB is used to solve this state equation. The computer configuration is as follows: Inter (R) Core (TM) i5-3317 U CPU $^@$ 1.70 GHz 1.70 GHz, Memory 3.86GB. Taking a belt conveyor at the Tongnu bulk cargo terminal of Nantong Tiansheng Gang Power Generation Co. LTD in China for example, the real parameters and simulation parameters are listed in Table 3. As described in Section 3.1, the conveyor belt at bearing segment is discretized in a very high resolution to model the changes of dynamic tension. Due to the length of belt conveyor is 4500 m and the belt speed is 3.75 m/s, 15 m as an appropriate for the unit lengths of belt at bearing segment if *T* = 4 s. The unit lengths of belt at return segment set 30 m for roughly discretized to reduce computational complexity. The value of rheological constant $\tau$ is constant, see in description of formula (10). The conveyor belt stiffness coefficient *k* is the unit length of $k_{tT}$, which is obtained from $k_{tT}$ (formula (7)) divided by *T*. Then multiply the conveyor belt stiffness coefficient *k* by rheological constant $\tau$ to get the damping coefficient of conveyor belt *c*. The coefficient independent of belt speed $C_{vo}$, the coefficient related to belt speed $C'_v$, the unit mass of conveyor belt $q_B$, the equivalent mass of upper idler $q_{RO}$, and the equivalent mass of lower idler $q_{RU}$ are obtained from the real parameters of the belt conveyor system.

The Harrison sinusoidal acceleration curve [49] is selected to apply a soft start control on the belt conveyor. In addition, to analyze the changing rules of the dynamic behavior of the conveyor belt under the non-uniform bulk material load, three load levels, i.e., empty-load, light-load, and over-load, are set. If the non-uniform bulk material distributions on the Tongnu bulk conveyor system are measured in the field test, we could obtain more accurate simulation results of dynamic behavior of the conveyor system. However, the dynamic characteristics of a conveyor belt is so complicated that the non-uniform bulk material distribution on belt. It may cause tremendous economic loss to bulk terminal if major security incidents accidents happened, such as strip breakage or longitudinal tear. Therefore, for the safety and experimental feasibility, we perform the simulation first by setting the corresponding instantaneous material flow parameters (shown in Table 4) to analyze the belt mechanical behavior under non-uniform bulk material distribution on the belt. In order to measure the non-uniform bulk material distributions on the moving belt at speed of 3.75 m/s in the laboratory, the iron ore powder was used for the measurement material and it was poured on the belt artificially

in pre-set flow rates from the guild chute. Then substituting the non-uniform distributions of the iron ore powder measured by the experimental facility into the matrix of *M*, *C*, *K* in Equation (21), the belt tensions, accelerations, and displacements at the head and tail of the belt conveyor system are easily obtained by simulation.

**Table 3.** Real and simulated parameters of the belt conveyor.

| Real Parameters | Numerical Value | Simulated Parameters | Numerical Value |
|---|---|---|---|
| Length of belt conveyor, $L$ (m) | 4500 | Unit length of belt at bearing segment, $l$ (m) | 15 |
| Bandwidth, $B$ (mm) | 1600 | Unit length of belt at return segment, $l$ (m) | 30 |
| Belt speed, $v$ (m/s) | 3.75 | Coefficient related to belt speed $C'_v$ | 0.002 |
| Conveying inclination, (°) | 8° | Coefficient independent of belt speed $C_{vo}$ | 0.025 |
| Upper idler spacing, (m) | 1.5 | Rheological constant, $\tau$ | 0.8 |
| Lower idler spacing, (m) | 3 | Conveyor belt stiffness coefficient, $k$ (kN/m) | 4267 |
| Elastic modulus, $E_B$ (kN/mm) | 160 | Damping coefficient of conveyor belt, $c$ (kN/m) | 3414 |
| Diameter of all idlers, (mm) | 159 | Unit mass of conveyor belt, $q_B$ (kg/m) | 53.472 |
| Length between tensioning device and head drum, (m) | 100 | Equivalent mass of upper idler, $q_{RO}$ (kg) | 26.18 |
| Number of element, $n$ | 450 | Equivalent mass of lower idler, $q_{RU}$ (kg) | 19.22 |

**Table 4.** Section and corresponding load of the belt conveyor.

| | Subsection | 1–69 | 70–129 | 130–189 | 190–249 | 250–300 |
|---|---|---|---|---|---|---|
| Empty-load | $q_{G1}$/kg/s | 0 | 0 | 0 | 0 | 0 |
| Light-load | $q_{G2}$/kg/s | 300 | 280 | 270 | 290 | 260 |
| Over-load | $q_{G3}$/kg/s | 600 | 680 | 670 | 690 | 660 |

## 5. Results and Discussion

### 5.1. Influence of Bulk Material Distribution on Start-Up Dynamic Characteristics

Starting under Harrison sinusoidal acceleration, the starting time *T* is 50 s, the maximum band speed under stable operations is 3.75 m/s, and the simulation time is 50 s. Then, the dynamic characteristic surface of the belt conveyor under empty-load, light-load and over-load is shown in Figure 10.

In Figure 10, it can be seen from the tension-time history of all elements of the conveyor belt that during the starting process of the belt conveyor, the head needs to overcome the inertia of the body and the friction resistance of the system, which generates strong tension fluctuations. In particular, under the over-load, the influence on the tension of the conveyor belt is obvious, and this tension lasts for a long time. However, due to the length of the belt conveyor, the transmission of tension needs time to overcome the resistance along the way, so the amplitude of belt tension fluctuates after the number of segments decreases, which is consistent with the realistic tension distribution law of the belt conveyor. It also shows that the conveyor belt starts to accelerate at approximately 12 s at the initial section. The speed curve turns smooth after about 25 s and the belt speed reaches 3.75 m/s at 50 s,

which is in line with the starting speed curve. From the time history of all unit displacements of the conveyor belt, it can be seen that the displacement of the conveyor belt keeps growing with increasing time and that the segment terminal of the conveyor belt fluctuates greatly.

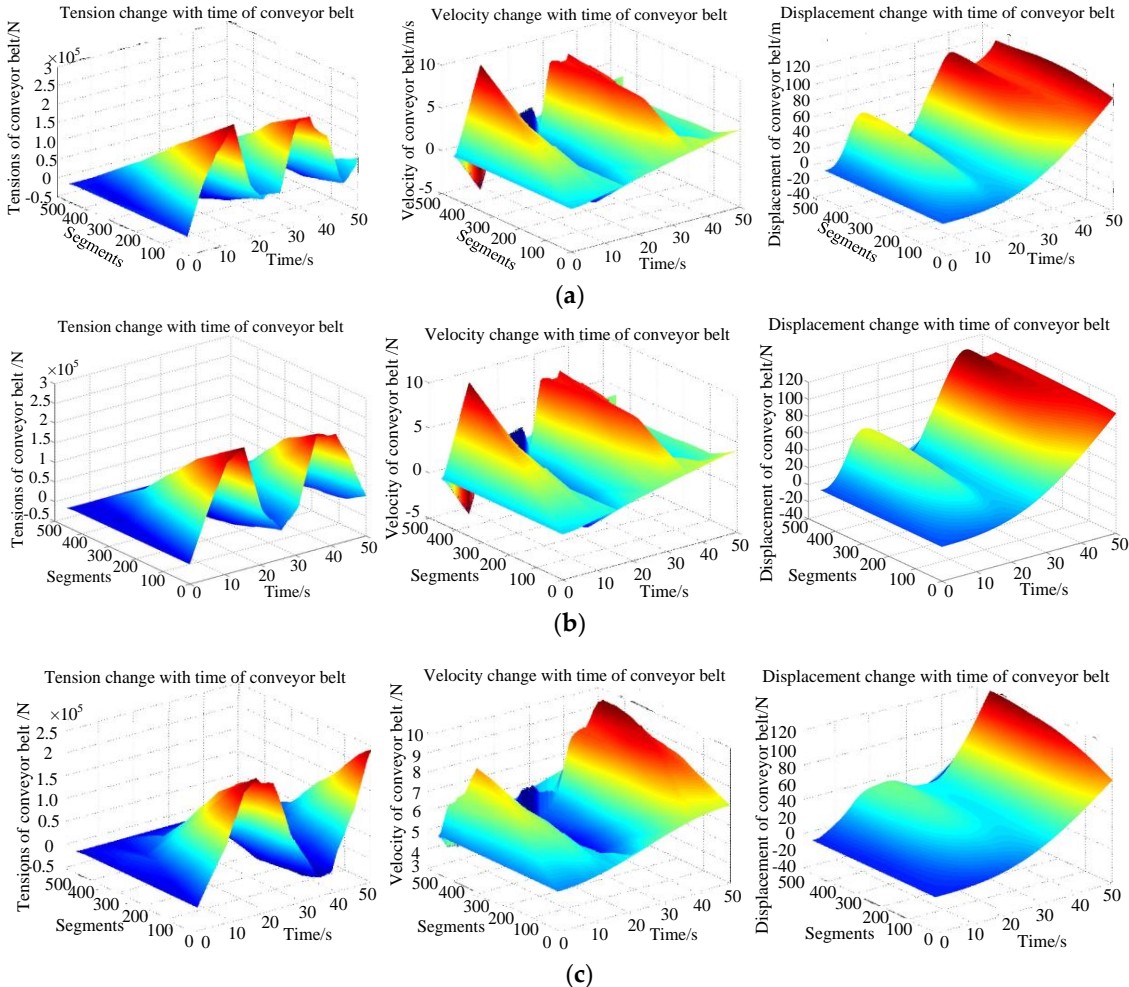

**Figure 10.** Dynamic characteristic curves of the belt conveyor. (**a**) Empty-load. (**b**) Light-load. (**c**) Over-load.

## 5.2. The Influence of Starting Time on the Dynamic Characteristics of the Belt Conveyor

The Harrison sinusoidal acceleration control is adopted in the belt conveyor under light-load, and the starting time is set at 50 s, 100 s, and 150 s. The dynamic characteristics of the conveyor belt are shown in Figure 11.

In Figure 11a, the soft-start control curve is the same, however the starting time is different. The longer the starting time, the less the starting acceleration of the conveyor belt. It is found that the probability of peak dynamic tension in the conveyor belt at the head and the tail of the machine is significantly reduced. It can be seen from the figures that the maximum tension on the conveyor belt presents a law of exponential decline as the starting time increases. When the starting time is less than 100 s, the peak value of the dynamic tension of the conveyor belt increases significantly, and the conveyor belt vibration is more severe. After prolonging the starting time, the dynamic characteristics of the conveyor belt are obviously improved.

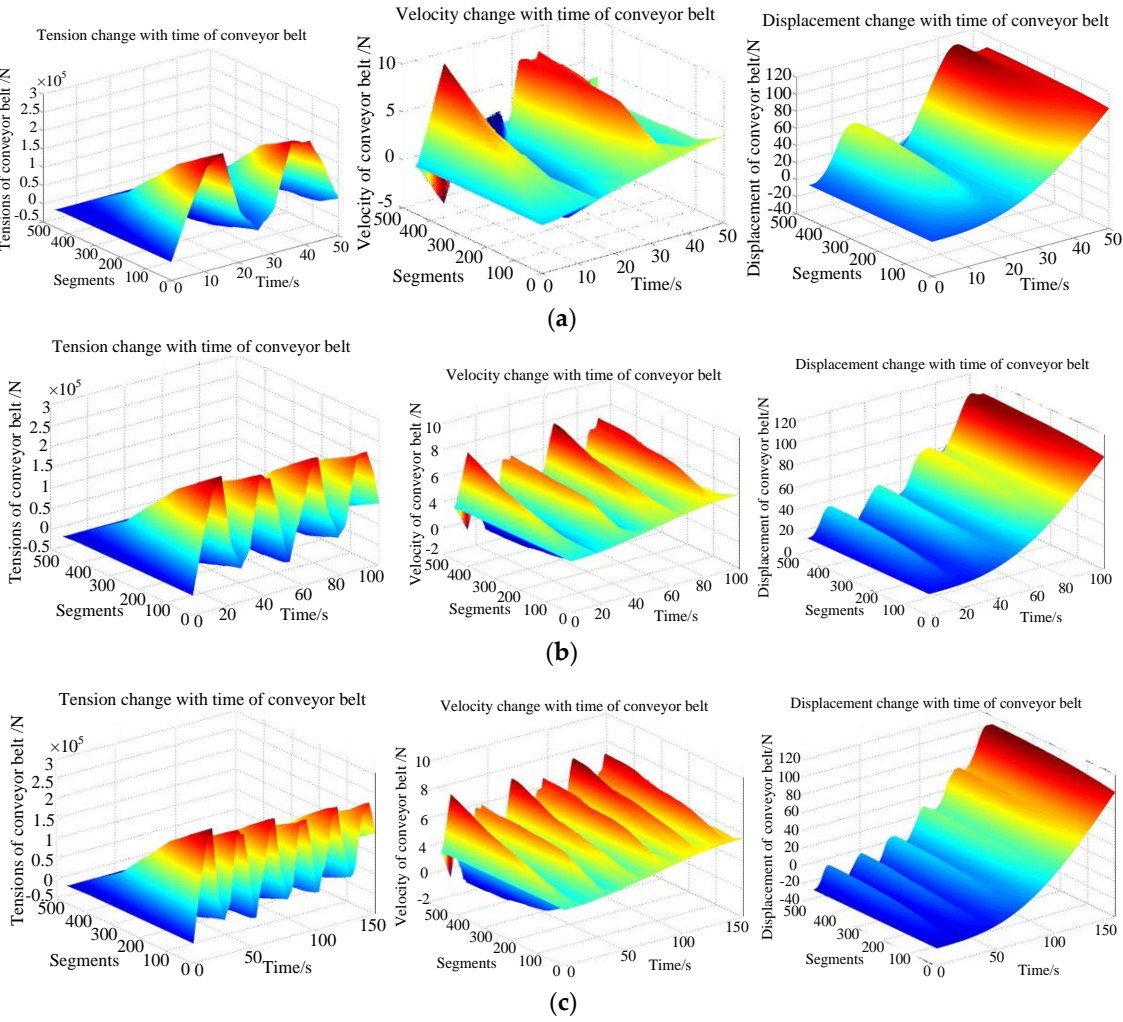

**Figure 11.** Dynamic characteristic curves of the belt conveyor under different starting times. (**a**) Starting time is 50 s. (**b**) Starting time 100 s. (**c**) Starting time 150 s.

### 5.3. Utility for Energy Efficiency Improvement

In speed control, the optimal speed regulation time should be chosen for a good dynamic performance of the belt conveyor system. In this way, the risks, such as belt overloading, belt slippage, and belt tear or breaking can be prevented. Based on presented results, the dynamic model of a belt conveyor considering the non-uniform bulk material distribution would be very useful for optimization of the speed regulation time of the belt conveyor. It should be pointed out that this dynamic model is capable of refined calculating the belt tensions of all elements of the belt conveyor system with the bulk material distribution, especially the belt tensions around the drive pulley. With the dynamic model, simulations can be carried out to analyze the dynamic characteristics of the belt conveyor during the regulation operation if the initial regulation time is obtained. In this way, the risks can be predicted in advance. In addition, more simulations should be carried out to get the optimal regulation time to improve the regulation performance.

Generally, a series of influencing factors, such as external constraints (electricity consumption, electricity cost, greenhouse gas emissions, etc.), operation efficiency (throughput rate, average throughput, etc.), and system constraints (the belt, idler, driver system, etc.), should be considered in speed control strategies for the energy efficiency improvements. However, the relationship among them is hardly expressed by a mathematic function. Therefore, there are a couple of negative side effects of a speed control strategy that solely or primarily focus on equipment level or operational

level. Most of the existing strategies make belt conveyors operate at unsuitable operating points since they do not consider the system constraints and external constraints. There must be a mechanism to compensate the inaccuracy caused by modelling uncertainties, external disturbances, and unexpected reactions of relevant components. However, during the process of conveying bulk materials on a belt conveyor, any belt speed changes will lead to nonlinear changes in the drive tension of the belt. In particular, if the non-uniform bulk material distribution is exerted a sudden dynamic shock on the belt, the tension of the belt will change greatly, which may lead to skidding, damage or even tearing. In actual situations, this may lead to misjudgment, affecting safety. We know that the speed optimization can play an important role in the energy efficiency improvement of the belt conveyor. Energy efficiency improvement brings not only high efficiency and low electricity consumption but also high safety. Therefore, developing a reasonable speed control strategy is the key for energy savings. The dynamic characteristics of the belt should be considered in the construction of the speed control strategy of the belt conveyor.

## 6. Conclusions

The characteristics of a long conveyor belt is complicated when it involves the interactions of the belt and the non-uniform bulk material distribution. It influences the speed control technology for energy efficiency significantly, especially for the conveyor belt applications. Therefore, for the speed control, it is necessary to establish a high-precision dynamic model to analyze the dynamic performance of a conveyor belt. However, the current dynamic belt models typically assume that the material distribution on the belt is uniform. The effects of the non-uniform bulk material distribution on the belt tension are rarely considered. By implementing the laser scanning technology, a non-uniform bulk material distribution model is firstly proposed. The model is verified by the experiment in realistic operational conditions. Then, a high-precision longitudinal dynamic model of a belt conveyor considering the micro-units of actual load on the belt is proposed. FEM and MATLAB simulation platforms are used to form the dynamic modelling and simulation research of belt conveyors with non-uniform bulk material distribution. In the model, the measurement accuracy is less affected by the unevenly distributed and intermittent bulk materials. The dynamic behavior with non-uniform bulk material distribution, start time and conveyor belt position are studied comprehensively. The model is suitable for analyzing mechanical behavior and optimizing the operating procedures of belt conveyor systems. Further research will be focused on developing a speed control strategy considering the control objectives and the dynamic characteristics of the system comprehensively.

**Author Contributions:** Conceptualization, F.Z. and Q.W.; methodology, F.Z.; formal analysis, T.W.; validation, C.Y. investigation, C.Y. and F.Z.; writing—original draft preparation, F.Z.; writing—review and editing, T.W. and F.Z.; funding acquisition, F.Z. All authors have read and agreed to the published version of the manuscript.

**Funding:** This research was funded by the National Natural Science Foundation of China under Grant number 61703215, the Ministry of Transport and applied basic research project of China under Grant number 2013329811340, and the National Defense Pre-Research Project of Wuhan University of Science and Technology under Grant number GF201809.

**Conflicts of Interest:** The authors declare no conflict of interest.

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
