# Peer review of "Dynamic Behaviour of a Conveyor Belt Considering Non-Uniform Bulk Material Distribution for Speed Control"

_applsci, doi:10.3390/app10134436_

Round 1

Reviewer 1 Report

1. In FIG. 7 there is an error, because point 7 cannot be in the middle of the length of the conveyor belt due to the inclination of the conveyor line

2. It is necessary to explain difference in meaning of the symbol t and symbol tk. If in tT we have - “t is the t-th unit of time” (line 212), then what is tk (line 211 and next).

3. There is an error on line 359 because equation 21 is not the result of the conversion of equation 16

4. There is an error in the article between the link numbers in the text and their numbers in REFERENCES section

5. There is an error in REFERENCES section, the article Statkic S. et al. is listed twice.

6. Аuthors make mistake when they don’t see the difference between term “heavy-load” (line 398) and term “overload” (table 3).

7. Section 4 “Conditions and modeling procedures” is very concise (36 lines total) and leaves a number of key aspects open. In particular:

7.1 It is not explained what technical means were used to obtain the data shown in Fig. 8

7.2 The meaning of the last phrases of section 4.1 is not clear enough (lines 385-387)

7.3 It is not disclosed how the data presented in table 2 are obtained.

8. The presentation in the "Results and discussion" section (5.3) is overloaded with non-constructive general reasoning.

9. The authors did not investigate the problem of energy efficiency, therefore, the corresponding phrase in the conclusion misleads the reader (line 480).

10. The article does not have enough data from real practice to confidently state that “The model is verified by the experiment in realistic operational conditions” (line 490)

11. It seems that the following conclusions key phrase require correction. In line 492 we read “The model is capable of analyzing mechanical properties”, but it should be - The model is suitable for analyzing mechanical behavior.

Author Response

Response to Reviewer 1 Comments

Dear Reviewer:

We appreciate very much the referee’s time and effort to review our paper and the in-depth and valuable comments. We have considered these comments and incorporated them fully into the revised manuscript, which, we believe, has improved the quality of the paper.

Point 1: In FIG. 7 there is an error, because point 7 cannot be in the middle of the length of the conveyor belt due to the inclination of the conveyor line.

 Response 1: We agree with the reviewer. The inclination of the conveyor line impacts the position of barycentre on belt between two adjacent roller sets directly.

 Action 1: We have corrected our mistake in Fig. 7 in the revised manuscript.

 Point 2: It is necessary to explain difference in meaning of the symbol t and symbol tk. If in tT we have - “t is the t-th unit of time” (line 212), then what is tk (line 211 and next).

 Response 2: This is indeed a very detailed review and valuable suggestion. In the section 2.3, T means the unit of time, which contains T seconds; t is the t-th unit of time, which contains tT seconds; k is the number of material flow cross-section areas per unit time. Then tk means the k frame of t-th unit of time. Due to the scanning frequency of the laser scanner is fspeed (frames per second), k is obtained by k=T·fspeed.. Compared with the tT (t is from 1~t), the tk(t is from 1~t, k is from 1~k) can much better reflect the original time series.

 Action 2: In the revised manuscript, we add a more detailed description about this.

 Point 3: There is an error on line 359 because equation 21 is not the result of the conversion of equation 16.

 Response 3: We agree with the reviewer. On line 359, equation 21 is the conversion of equation 20.

Action 3: In the revised manuscript, we have made a correction according to the Reviewer’s comments.

 Point 4: There is an error in the article between the link numbers in the text and their numbers in REFERENCES section.

 Response 4: Agreed and is a valuable suggestion. We have re-examined the original references section and found that we made a mistake of missing number in reference “Gladysiewicz, L.; Robert Krol,; Kisielewski W. Measurements of loads on belt conv....”, which makes all the other references irrelevant to the text.

 Action 4: We have corrected our mistake in the revised manuscript. And every reference cited in the text has been checked carefully.

 Point 5: There is an error in REFERENCES section, the article Statkic S. et al. is listed twice.

 Response 5: Agreed and is a very valuable suggestion for us to improve the quality of the paper. We have made a mistake of listing the article Statkic S. et al. twice in Ref. 5 and 45 in original manuscript, respectively.

 Action 5: We have deleted the article Statkic S. et al.in Ref. 5 and added a reference to link the [4] in the text.

 Point 6: Аuthors make mistake when they don’t see the difference between term “heavy-load” (line 398) and term “overload” (table 3).

 Response 6: This is indeed a very detailed review and valuable suggestion. We have re-examined the original manuscript and found that we made a mistake. The three load levels of non-uniform bulk material are empty-load, light-load and over-load respectively.

Action 6: We have corrected our mistake to make sure these terms are consistent with the table 3 in the revised manuscript.

Point 7: Section 4 “Conditions and modeling procedures” is very concise (36 lines total) and leaves a number of key aspects open. In particular:

 7.1 It is not explained what technical means were used to obtain the data shown in Fig. 8.

 Response 7.1: This is indeed a very insightful and valuable suggestion. In our previous study (Zeng, F.; Wu, Q.; Chu, X.; Yue, Z. Measurement of bulk material flow based on laser scanning technology for the energy efficiency improvement of belt conveyors. Measurement. 2015, 75, 230-243.), we described the design and verification of the system based on laser scanning technology in detail. We did not feel it necessary to repeat this information in this paper. Thus, we describe the measurement facility briefly and propose a model of bulk material distribution in the section 2. To measure the distribution of the bulk material on the moving belt, a suitable belt speed for material transportation is required. The belt speed needed to be stable over the time period of the start-up process. Speed of 0.5 m/s, 1.0 m/s and 1.5 m/s are selected as suitable values, which are controlled using the frequency converter. Standard volumes of 1 L, 3 L, 5 L are then used to verify the volumes measured by the instrument. Fig. 8 (a) shows the measured instantaneous bulk material flow distribution with an artificial loading instantaneous bulk material flow rate of 8.911 kg/s (3 L/s).  These data are obtained at a stable speed of 1.0 m/s. Then a mean filter is adopted to smooth the original data. Figure 8(b) shows the result of comparing the average value obtained from the actual detection value of every 30 frames with the expected flow.

Action 7.1: We have added a clear explanation about the data acquisition means in Section 4.1.

 7.2 The meaning of the last phrases of section 4.1 is not clear enough (lines 385-387)

 Response 7.2: We agree with the reviewer.

Action 7.2: In the revised manuscript, we modify these phases and add a more detailed description in section 4.1. Some analysis about the results of the measurement of bulk material distribution has been added. Besides, more details have been added in proving that the accurate bulk material distribution is a prerequisite of complicated dynamic modelling of a belt conveyor system. Particularly, the analysis of the belt mechanics is important for researching belt dynamic characteristics, which has effect on its operation factors decision making.

 7.3 It is not disclosed how the data presented in table 2 are obtained.

 Response 7.3: We agree with the reviewer. The left data presented in table 2 are the real parameters of a belt conveyor system, which are obtained from Tongnu bulk cargo terminal of Nantong Tiansheng Gang Power Generation Co. LTD in China. The right of the table lists the simulation parameters for simulation.

 Action 7.3: In the revised manuscript, we have added a clear explanation about the data sources in Section 4.2.

 Point 8: The presentation in the "Results and discussion" section (5.3) is overloaded with non-constructive general reasoning.

 Response 8: This is a good suggestion. As mentioned in the literature review, traditional strategies for improving energy efficiency of belt conveyors had rarely considered the non-uniform bulk material distribution. In fact, the complicated dynamic characteristics of a belt conveyor system is a prerequisite of optimal speed control. Particularly, the analysis of the belt mechanics is important for researching belt dynamic characteristics, which has effect on its operation factors decision making. Thus the main purpose of this paper is to propose a high-precision dynamic model intended to determine the dynamic characteristics of a conveyor belt under non-uniform bulk material distribution. It will contribute to the establishment of speed control constraints for developing a control strategy for practical application. Section 5.3 presents the utility of the high-precision dynamic model for energy efficiency improvement, which is our future research on this topic.

Action 8: We change the title of section 5.3 to “Utility for energy efficiency improvement” in the revised manuscript. We will perform the utility of the high-precision dynamic model for energy efficiency improvement in the future study.

 Point 9: The authors did not investigate the problem of energy efficiency, therefore, the corresponding phrase in the conclusion misleads the reader (line 480).

 Response 9: This is indeed an insightful suggestion. As mentioned in the introduction, the bulk material flow transported by a belt conveyor is variable in a large number of cases. And the belt can be only partially filled when the flow rate is smaller than the nominal conveying capacity. Accordingly, the instantaneous cross-section area of material on the belt is usually less than the rated value, which leads to the great waste of electricity energy. Additionally, uneven distribution of material along the length of belt certainly affects the driver power. Therefore, if the speed is changed to maximize the volumetric capacity with the varying supplied capacity, the energy efficiency of belt conveyors could be improved. Since the instrument has the ability to acquire the cross-section area of material which is deposited onto the belt, we can speculate that the control strategy will be implemented in practical according to the criterion of optimum utilization efficiency of the transported material cross section. With this in mind, and considering the operation of the belt with reduce capacity, it is clear that significant energy savings can be achieved.

 Action 9: The reviewer’s suggestion motivates us to investigate the energy efficiency of belt conveyors in the future work.

Point 10: The article does not have enough data from real practice to confidently state that “The model is verified by the experiment in realistic operational conditions” (line 490)

 Response 10: This is indeed a very insightful suggestion. The authors designed and conducted experiments of the bulk material distribution measurement based on laser scanning technique, and found that the experimental results can better reflect the real state of material transportation. Therefore, in the subsequent field test, the bulk material distribution can be collected by this measurement system. However, the dynamic characteristics of a conveyor belt is so complicated that the non-uniform bulk material distribution on belt. It may cause tremendous economic loss if major security incidents accidents happened, such as strip breakage or longitudinal tear. Therefore, for the safety and experimental feasibility, we perform the simulation first to analyse the belt mechanical behaviour under non-uniform bulk material distribution on the belt. The results can contribute to optimize the operating procedures of belt conveyor systems.

Action 10: We have modified these phases for academic rigour in the revised manuscript. We will perform the mechanical behaviour field test as a future work to obtain more comprehensive study of this method.

Point 11: It seems that the following conclusions key phrase require correction. In line 492 we read “The model is capable of analyzing mechanical properties”, but it should be - The model is suitable for analyzing mechanical behavior.

 Response 11:We agree with the reviewer.

 Action 11:We have corrected this key phrase according to the reviewer’s suggestion in the revised paper.

Reviewer 2 Report

It is an interesting paper and deserves to be published. It's not free of language mistakes and minor flaws. The comparative analysis of the belts material distribution need an error analysis between the results. The authors should work on the conclusions and add numerical results which would support them. There are mistakes in the references like missing number and also incorporated in the text.

Minor flaws: The authors are using Simulink software (can be seen on figures of the test stand) for calculations which is incorporated into Matlab but there is not a lot about it in the text, which I accept as a minor flaw.

There are some minor flaws in the figures, which are still readable but should be corrected as in Figure 8(b) in the description of the y axis.

As for the missing error analysis, there are multiple possibilities to perform that, e.g. the authors could perform the analysis of variance of the belt measured instantaneous material flow and the ideal uniform distribution data and with the use of this method they could perform the actual error analysis between the results.

There are a lot of sentences where the authors have chosen a wrong tense or the sentence structure is wrong (that’s why I prosed Moderate English changes required) some examples which I remember:
The fluctuation of material flow on the conveyor belt will lead the detrimental vibrations..... -should be: ...will lead to...
...dynamic model is capable for refined.... -should be: ...is capable of...
.....conveyor is complicated when involves... -should be:...when it involves....

The conclusions should refer to numerical values which would allow the fast reader, which usually reads only the abstract and conclusions to gain the knowledge if the presented results will be interesting to him/her. The authors didn’t left a window open to speak about future research on this topic or other research ideas, which is important in the conclusions.

Some of the mistakes in references like missing number:
Gladysiewicz, L.; Robert Krol,; Kisielewski W. Measurements of loads on belt conv....
Which makes all the other references irrelevant to the text as You have to search for the appropriate reference.

Author Response

Response to Reviewer 2 Comments

Dear Reviewer:

We appreciate very much the referee’s time and effort to review our paper and the in-depth and valuable comments. We have considered these comments and incorporated them fully into the revised manuscript, which, we believe, has improved the quality of the paper.

Point 1: It is an interesting paper and deserves to be published. It's not free of language mistakes and minor flaws. The comparative analysis of the belt’s material distribution need an error analysis between the results. The authors should work on the conclusions and add numerical results which would support them. There are mistakes in the references like missing number and also incorporated in the text.

Response 1: This is indeed a very detailed review and valuable suggestion. According to our previous study (Zeng, F.; Wu, Q.; Chu, X.; Yue, Z. Measurement of bulk material flow based on laser scanning technology for the energy efficiency improvement of belt conveyors. Measurement. 2015, 75, 230-243.), we described the design and verification of the system based on laser scanning technology in detail. Therefore, in the paper we just describe the measurement facility briefly and propose a model of bulk material distribution in the section 2. Besides, we have re-examined the original references section and found that we made a mistake of missing number in reference “Gladysiewicz, L.; Robert Krol,; Kisielewski W. Measurements of loads on belt conv....”, which makes all the other references irrelevant to the text.

Action 1: The authors are now performing the numerical research about this aspect. The reviewer’s suggestion motivates us to do more in-depth research. In the revised manuscript, we have added a clear explanation about the data acquisition means and an error analysis to support the results in Section 4.1. In addition, we have corrected our mistakes of language and references. Every reference cited in the text has been checked carefully.

Point 2: Minor flaws: The authors are using Simulink software (can be seen on figures of the test stand) for calculations which is incorporated into Matlab but there is not a lot about it in the text, which I accept as a minor flaw.

Response 2: This is indeed an insightful suggestion. In section 5, all the simulation cases are coded with M files in Matlab R2010b. The user interface is designed by the graphical user interface (GUI) toolbox of Matlab. When the bulk material distribution measured by the experimental facility mentioned in section 2.2, the real and simulated parameters of the belt conveyorare input to the GUI, the matrix of M, C, K in equation (21) can be built. Weselect the Harrison sinusoidal acceleration curve to apply a soft start control on the belt conveyor. After that, the ode 45 function in Matlab is used to solve this state equation. The computer configuration is as follows: Inter (R) Core (TM) i5-3317 U CPU @ 1.70 GHz 1.70 GHz, Memory 3.86GB.

Action 2: In the revised manuscript, related content has been added in section 4.2.

Point 3: There are some minor flaws in the figures, which are still readable but should be corrected as in Figure 8(b) in the description of the y axis.

Response 3: Agreed and is a very valuable suggestion for us to improve the quality of the paper.

Action 3: We have corrected these figures in the revised manuscript.

Point 4: As for the missing error analysis, there are multiple possibilities to perform that, e.g. the authors could perform the analysis of variance of the belt measured instantaneous material flow and the ideal uniform distribution data and with the use of this method they could perform the actual error analysis between the results.

Response 4: This is indeed a very insightful and valuable suggestion.

Action 4: The reviewer’s suggestion motivates us to do more in-depth research. In the revised manuscript, we have added an error analysis to support the results in Section 4.1.

Point 5:There are a lot of sentences where the authors have chosen a wrong tense or the sentence structure is wrong (that’s why I proposed Moderate English changes required) some examples which I remember:

The fluctuation of material flow on the conveyor belt will lead the detrimental vibrations..... -should be: ...will lead to......dynamic model is capable for refined.... -should be: ...is capable of........conveyor is complicated when involves... -should be:...when it involves....

Response 5: This is indeed a very insightful and valuable suggestion.

Action5: We have corrected our mistake in the revised manuscript. And we have checked the tense and the sentence structure in the revised manuscript carefully.

Point 6: The conclusions should refer to numerical values which would allow the fast reader, which usually reads only the abstract and conclusions to gain the knowledge if the presented results will be interesting to him/her. The authors didn’t left a window open to speak about future research on this topic or other research ideas, which is important in the conclusions.

Response6: This is indeed a very insightful and valuable suggestion. As mentioned in the literature review, traditional strategies for belt conveyors’ energy efficiency improvement had rarely considered the dynamic characteristics of a belt conveyor in transient operation. In fact, due to the nonlinear and viscoelastic of a fabric conveyor belt, belt slipping around pulley and belt tearing at the splicing area may be happened if the speed adjustment is not properly controlled on the belt, especially when the speed changes dramatically. However, the theoretical calculation of longitudinal belt characteristics only considers a uniform bulk material distribution. Thus the main purpose of this paper is to propose a high-precision dynamic model intended to determine the dynamic characteristics of a conveyor belt under non-uniform bulk material distribution. The authors designed and conducted experiments of the bulk material distribution measurement based on laser scanning technique, and found that the experimental results can better reflect the actual load distribution on a conveyor belt. Based on this, a high-precision longitudinal dynamic model to investigate the dynamic behaviour of a belt conveyor is investigated. We represent the experimental verification results for the non-uniform bulk material distribution model and establish the discrete simulation model of the belt conveyor system using Matlab software. For the safety and experimental feasibility, we perform the simulation to analyse the belt mechanical behaviour under non-uniform bulk material distribution on the belt. We present a simulation test on the different belt tensions, accelerations and tensioning device displacements at the head and tail of the belt conveyor system. The discussion on the influence of the non-uniform bulk material distribution and starting time on the dynamic characteristics of the belt conveyor is proposed, which contribute to the establishment of speed control constraints for developing a control strategy for practical application. It will contribute to the establishment of speed control constraints for developing a control strategy for practical application. Section 5.3 presents the utility of the high-precision dynamic model for energy efficiency improvement, which is our future research on this topic.

Action6: We change the title of section 5.3 to “Utility for energy efficiency improvement” in the revised manuscript. We will perform the utility of the high-precision dynamic model for energy efficiency improvement in the future study.

Point 7:Some of the mistakes in references like missing number:

Gladysiewicz, L.; Robert Krol,;Kisielewski W. Measurements of loads on belt conv....

Which makes all the other references irrelevant to the text as you have to search for the appropriate reference.

Response7: This is indeed a very detailed review and valuable suggestion.

Action 7: In the revised manuscript, we have added a sequence number for the reference of “Gladysiewicz, L.; Robert Krol,;Kisielewski W. Measurements of loads on belt conv....”. And every reference cited in the text has been checked carefully.

Round 2

Reviewer 1 Report

In the first review it was already noted that Section 4 is very concise and leaves a number of key aspects open.

In particular - It is not disclosed how the «Simulated parameters» presented in table 3 (in v.1 it was table 2) are obtained. It is quite obvious that the correct determination of these parameters is very important to the adequacy of the simulation results.

At the same time, in section 4.2, the authors should explain how applicable are the results of laboratory measurements using laser scanning technology (section 4.1) when modeling the «Tongnu bulk cargo terminal» conveyor.

Author Response

Response to Reviewer 1 Comments

Dear Reviewer:

We appreciate very much the referee’s time and effort to review our paper and the in-depth and valuable comments. We have considered these comments and incorporated them fully into the revised manuscript, which, we believe, has improved the quality of the paper.

Point 1: In the first review it was already noted that Section 4 is very concise and leaves a number of key aspects open.

1.1 In particular - It is not disclosed how the «Simulated parameters» presented in table 3 (in v.1 it was table 2) are obtained. It is quite obvious that the correct determination of these parameters is very important to the adequacy of the simulation results.

Response 1.1: We agree with the reviewer. We list the real parameters of a belt conveyor system in the left of the table 3, which are obtained from Tongnu bulk cargo terminal of Nantong Tiansheng Gang Power Generation Co. LTD in China. For analyzing the dynamic behaviour of the belt conveyor system, the simulation parameters based on the real parameters are listed in right of the table 3. The first parameter is the unit lengths of belt at bearing segment, which is used for dividing the conveyor belt at bearing segment in a very high resolution to model the changes of dynamic tension. Due to the length of belt conveyor is 4500 m and the belt speed is 3.75 m/s, 15 m as an appropriate unit lengths of belt at bearing segment (section 3.1 , T=4 s). The unit lengths of belt at return segment set 30 m for roughly discretized to reduce computational complexity. The value of rheological constant τ is constant, see in description of formula (10). The conveyor belt stiffness coefficient k is the unit length of ktT, which is obtained from ktT (formula (7)) divided by T. Then multiply the conveyor belt stiffness coefficient k by rheological constant τ to get the damping coefficient of conveyor belt c. The coefficient independent of belt speed Cvo, the coefficient related to belt speed C′v, the unit mass of conveyor belt qB, the equivalent mass of upper idler qRO, and the equivalent mass of lower idler qRU are obtained from the real parameters of the belt conveyor system.

Action 1.1: We have added a clear explanation about the data sources of simulated parameters in Section 4.2.

1.2 At the same time, in section 4.2, the authors should explain how applicable are the results of laboratory measurements using laser scanning technology (section 4.1) when modeling the «Tongnu bulk cargo terminal» conveyor.

Response 1.2: This is indeed a very detailed review and valuable suggestion. If the non-uniform bulk material distributions on the Tongnu bulk conveyor system are measured in the field test, we could obtain more accurate simulation results of dynamic behaviour of the conveyor system. However, the dynamic characteristics of a conveyor belt is so complicated that the non-uniform bulk material distribution on belt. It may cause tremendous economic loss to bulk terminal if major security incidents accidents happened, such as strip breakage or longitudinal tear. Therefore, for the safety and experimental feasibility, we perform the simulation first by setting the corresponding instantaneous material flow parameters (shown in Table 4) to analyse the belt mechanical behaviour under non-uniform bulk material distribution on the belt. In order to measure the non-uniform bulk material distributions on the moving belt at speed of 3.75 m/s in the laboratory, the iron ore powder was used for the measurement material and it was poured on the belt artificially in pre-set flow rates from the guild chute. Then substituting the non-uniform distributions of the iron ore powder measured by the experimental facility into the matrix of M, C, K in equation (21), the belt tensions, accelerations and displacements at the head and tail of the belt conveyor system were easily obtained by simulation.

Action 1.2: In the revised manuscript, we add a more detailed description about the application of the results of laboratory measurements using laser scanning technology in Section 4.2. We will perform the field test as a future work to obtain more comprehensive study of this method.

Besides, in the revised manuscript, we tried our best to improve the manuscript and made some changes in the manuscript. These changes will not influence the content and framework of the paper. And here we did not list the changes but highlighted in yellow in revised paper. Finally, we still want to thank the anonymous referees for your hard work which has greatly improved the original manuscript, and hope that the correction will meet with approval.

Once again, thank you very much for your comments and suggestions.

Your sincerely

Fei Zeng, Cheng Yan, Qing Wu and Tao Wang

Reviewer 2 Report

The authors have corrected the paper according to the review. The highlighted mistakes are no longer present. Therefore i suggest to accept the paper in present form.

Author Response

Response to Reviewer 2 Comments

Dear Reviewer:

We appreciate very much the referee’s time and effort to review our paper and the in-depth and valuable comments. We have considered these comments and incorporated them fully into the revised manuscript, which, we believe, has improved the quality of the paper.

Point 1: The authors have corrected the paper according to the review. The highlighted mistakes are no longer present. Therefore i suggest to accept the paper in present form.

Response 1: In the revised manuscript, we have checked the English language and style carefully. We tried our best to improve the manuscript and made some changes in the manuscript. These changes will not influence the content and framework of the paper. And here we did not list the changes but highlighted in yellow in revised paper. Finally, we still want to thank the anonymous referee for your hard work which has greatly improved the original manuscript. Once again, thank you very much for your comments and suggestions.

Your sincerely

Fei Zeng, Cheng Yan, Qing Wu and Tao Wang
